# Training-Free Style and Content Transfer by Leveraging U-Net Skip Connections in Stable Diffusion

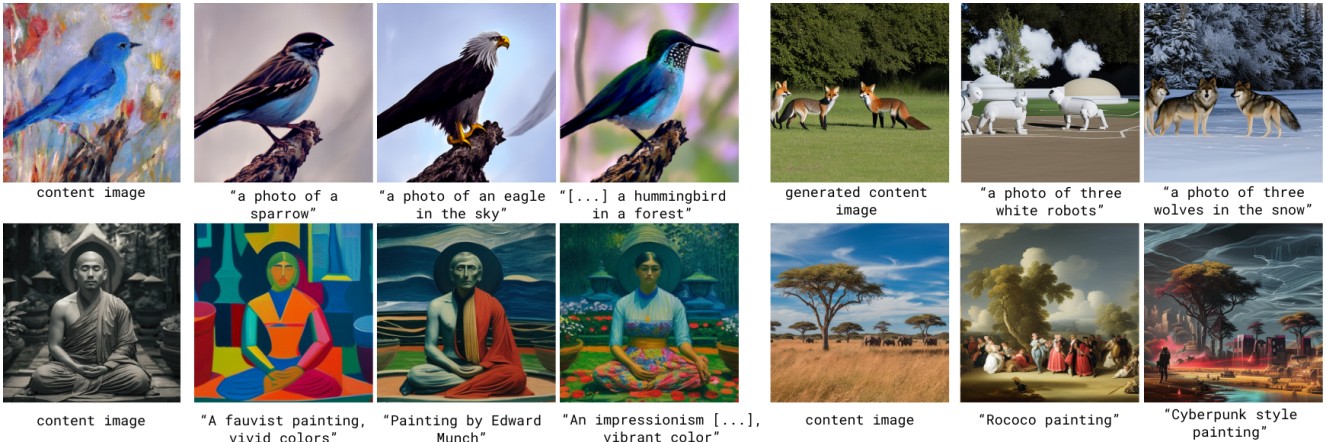

Figure 1. SkipInject: our method uses the l=4 and l=5 skip connections of Stable Diffusion to obtain flexible content and style transformations. From a painted "content image" of a bird, the model smoothly modifies the subject to resemble various species (e.g., sparrow, eagle) while retaining the overall scene. A generated image of foxes is transformed into "three white robots" and "three wolves in the snow,", with coherent and realistic alterations. Furthermore, the styles of the two content images are altered holistically, in aesthetics, subjects, and settings.

## Abstract

*Recent advances in diffusion models for image generation have led to detailed examinations of several components within the U-Net architecture for image editing. While previous studies have focused on the bottleneck layer (h-space), cross-attention, self-attention, and decoding layers, the overall role of the skip connections of the U-Net itself has not been specifically addressed. We conduct thorough analyses on the role of the skip connections and find that the residual connections passed by the third encoder block carry most of the spatial information of the reconstructed image, splitting the content from the style, passed by the remaining stream in the opposed decoding layer. We show that injecting the representations from this block can be used for text-based editing, precise modifications, and style transfer. We compare our method, SkipInject, to state-of-the-art style transfer and image editing methods and demonstrate that our method obtains the best content alignment and optimal structural preservation tradeoff.*

## 1. Introduction

Breakthroughs in diffusion models have unlocked unprecedented avenues for generating images and videos. Models such as Stable Diffusion [33], Midjourney, and Dall-E [31] have driven this evolution, with their outputs creating a transformative shift across diverse creative domains. Their influence reaches digital hobbyist circles, established professional practices like illustration, graphic design, and multimedia arts, and fosters innovative artistic exploration and community collaboration.

Despite the enormous generative affordances of these methods, broader output controllability is necessary for better adoption in creative communities, often reliant on a trial-and-error process of iterative refinement and on mood boarding and inspiration.

While previous generations of image generation mod-

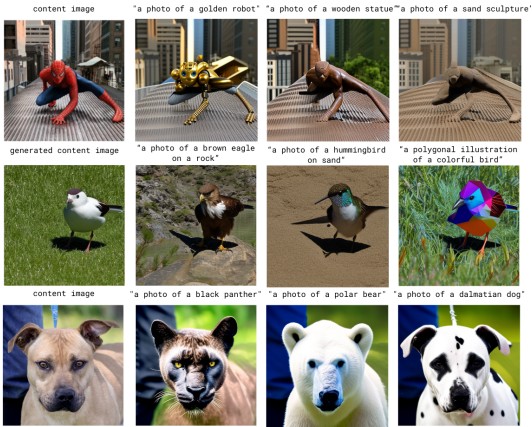

Figure 2. Examples of image editing results on Wild-TI2I and ImageNet-R-TI2I real and generated images.

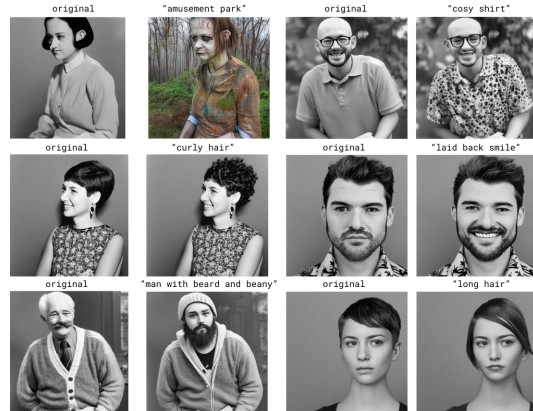

Figure 3. Image editing results on generated faces. We show precise transformations ranging from subtle changes, like makeup and hairstyle adjustments, to more global effects, including zombie-like effects. Our model preserves the core identity of each subject, maintaining facial structure.

els, including Variational Autoencoders [22] and Generative Adversarial Networks [20], leverage the latent space for image editing [14, 20, 37], diffusion models [16, 38] are based on a Markov chain denoising process and inherently lack a single latent space. In the context of U-Net-based diffusion models, training-free approaches to image editing focus on swapping different modules of the denoising architecture, including the self- and cross-attention modules and the h-space - the bottleneck of the U-Net. However, the skip connection - an essential element within the U-Net, aiding the transmission of long-range dependencies and the gradient propagation - has not been explored. In contrast to existing work, we focus on the former and its role in U-Net-based diffusion models.

To better understand the role of this module, we address the following questions: (i) How and where is information represented in the skip connections of the U-Net? (ii) How does it influence image generation? (iii) When does this information arise during the denoising process?

Interestingly, we observe that Stable Diffusion internally disentangles content from style within the third encoder/decoder block, with the content passing through the skip connection and the style through the main flow.

We find that injecting the third group of connections produced by the encoder from image $A$ to image $B$ transfers the spatial configuration of image $A$ onto image $B$. Conversely, we find that image $B$ transfers the style to image $A$ using the same injection, indicating that the corresponding third decoder block carries the style information. Additionally, leveraging the injection timestep controls the appearance of the background of image $B$ over image $A$, and modulating the mixing on the embedding offers control of the strength of the injection.

We demonstrate that an informed use of the properties of Stable Diffusion can achieve state-of-the-art performance on a wide variety of tasks, offering ample control over the intensity and nature of the output. In Sec. 5, we highlight the superiority of our method in achieving text-based image editing and style transfer and show preliminary results on fine-grained feature editing in Fig. 3.

To summarize, we contribute as follows:
- We investigate the role of the skip connections in the U-Net of Stable Diffusion, assessing their properties, their influence on the image, and variation across time steps.
- We propose an efficient and controllable image editing method and prove superiority or on-par SOTA performance on transferring content and style.
- Lastly, we propose three alternatives to modulate the editing effect.

## 2. Related work

In this section, we shortly explain the importance of latent space studies in the contexts of media studies and digital arts to further motivate the focus of this paper. Successively, we cover image editing methods on Stable Diffusion.

### 2.1. Latent space in the arts and humanities

The latent space, understood strictly as the space where the data lies in the bottleneck layer of a model, is a topical entity for studying and understanding models beyond technical fields. These spaces are studied as n-dimensional cultural objects [32]. The latent spaces make continuous and spatialized the cultural knowledge fed into or generated by the model, creating an implicit meaningful organization [36]. These representations can be then studied as a map of culture [43], and can, in turn, be used to study models as cultural snapshots of reality [7, 17, 44, 45].

Digital artists and creative industries extensively used latent space-rooted methodologies, such as latent space walks

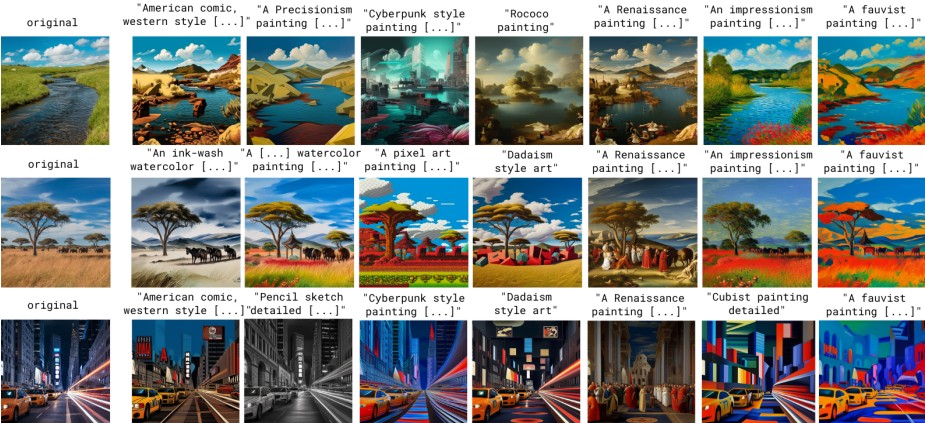

Figure 4. Examples of style transfer results on the Artist dataset [19].

and interpolation, to take advantage of the semantic continuity of this space. Initiating with DeepDream [2], the latent space continuity, opposed to reality's fragmentation, creates an attractive space of artistic hallucination [1].

## 2.2. Image manipulation

In this section, we present some of the pivotal works in this direction, organized by what element is used for editing.

**Latent code-based editing.** Asyrp [23] uses the *h-space* and CLIP supervision to find a direction of modification in the space at each timestep, to add to the original latent in the denoising process through a modified Diffusion Deterministic Implicit Model (DDIM) [39]. Boundary diffusion [48], on the other side, computes a modification direction that is injected only at the mixing step, testing both $\epsilon$-*space* and *h-space*. Haas *et al*. [11], among other findings, show that injecting the *h-space* of an image into another image changes the high-level semantics while retaining the structure and background. InjectFusion [18] observe the same phenomenon, implementing a calibrated procedure to inject the new *h-space*, maintaining the same correlation to the skip connections. These methods are mostly based on unconditional DDPM-based models trained on specific datasets for *e.g.* CelebA.

**Module-based editing.** Prompt2Prompt (P2P) [13] substitute the cross-attentions of the U-Net layers to obtain text-based image editing. Plug and Play (PnP) [42] find that accurate editing can be achieved by injecting the spatial features of the middle decoding and self-attention layers. Closely related to the two previous works, Liu *et al*. [24] investigate the role of the cross-attention and the self-attention in the different feature layers, observing again that intermediate features are the most salient. Finally, Artist [19] shows that using the middle residual blocks as PnP to control the content and the cross-attentions to inform the style obtains successful text-driven stylization.

**Text-based editing.** A common alternative for diffusion models leverages the manipulation of text conditioning. Methods like DiffusionCLIP [21] and InstructPix2Pix [4] fine-tune the model or the text conditioning to obtain desired edits. Various successful methods tackle personalizing the outputs to specific entities such as Dreambooth [35]. Lastly, methods like SDEdit [8] leverage partial inversion and text-guided generation to achieve fast, training-free editing.

**Adapters.** Other popular methods leverage adapters, including ControlNet [46] and T2IAdapter [27] to increase the modalities that can be used to control the diffusion process. In fact, they train an ad-hoc adapter for each additional modality, obtaining perceptually interesting outcomes. To increase the manipulability, other methods make use of specifically trained LoRA adapters, like PreciseControl [28] CTRLorALTer [40], LoRAdapter [10], which can achieve controlled modifications for the trained semantic.

## 2.3. Novelty

Compared to existing methods, our approach is the simplest but allows the greatest control, using numerous plug-ins to modulate the effect and allowing, in the same pipeline, editing the content and the style of the image. Lastly, we show that our method performs well on Turbo alternatives, obtaining the fastest results.

## 3. Preliminaries

In this section, we introduce Latent Diffusion Models (LDMs) as introduced by Rombach*et al*. [33], with a particular emphasis on its U-Net [34].

### 3.1. Latent Diffusion

Latent Diffusion Models (LDMs) overcome pixel-based diffusion models' high computational and memory costs by conducting the diffusion process in a reduced latent space.

To achieve this, a pre-trained autoencoder converts an image into a compact latent representation $z_0$ of $1/8$ the original per side size. The diffusion process is then applied to $z_0$, which substantially lowers the resource demands during training and sampling. During training, the model is optimized to predict the noise via a neural network, the U-Net.

## 3.2. Components of the U-Net

In our work, we leverage a pre-trained text-conditioned Latent Diffusion Model, which employs a U-Net backbone, popularly recognized as Stable Diffusion (versions 1.4, 1.5, 2, and 2.1). In Stable Diffusion, the conventional U-Net architecture [34] is enhanced with attention mechanisms, including self- and cross-attention blocks.

The **residual block** is inputted the latent features $\phi_t^{l-1}$ from the previous layer $l-1$ and outputs both the latent features to be inputted to the following block $\phi_t^l$, and the skip connections $f_t^l$ concatenated directly to the corresponding decoding layer, as:

$$f_t^l, \phi_t^l = \text{ResBlock}(\phi_t^{l-1}), \tag{1}$$

where ResBlock includes convolutional layers.

Stable Diffusion 1-2 models feature four encoding blocks, a bottleneck, and four mirroring decoding blocks. Each of the blocks contains three subblocks, each passing one skip connection. In the remainder of the paper, we refer to the skip connections by number, where $l = 0$ is the first skip connection and $l = 12$ is the one preceding the blottleneck.

## 4. Analysis

As explained in the previous section, skip connections are a critical component of the U-Net backbone, allowing long-range information flow and avoiding the vanishing gradient problem. However, their role within the Stable Diffusion models remains unknown. In this section, we present our investigation of the role of each skip connection, the time steps, and the properties of these embeddings to shed some light on these behaviors.

### 4.1. The role of skip connections

To analyze the effect of each skip connection, we store the skip connections of an injection image $A$ and test the injection of each skip connection and combinations of them into the original image $B$. We start from a common initial noise $z_t$. We follow two different noise selection strategies: (i) we randomly sample a $z_t$ from a Gaussian distribution, or (ii) we use the result of the DDIM inversion of either $A$, i.e., $z_t^A$, or $B$, i.e., $z_t^B$. We first fully denoise $z_t$ with the prompt of image $A$, $p^A$, and store the skip connection $f_t^l$ at each time-step $t$. Successively, we denoise $z_t$ using the prompt of image $B$, $p^B$. At each time-step from $t_{start}$ to $t_{end}$, we

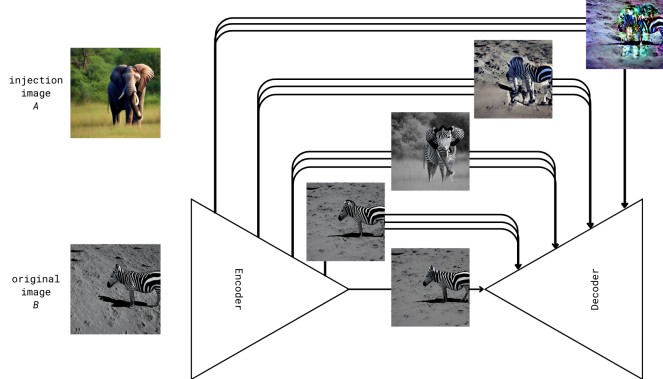

Figure 5. Visualization of the effect of switching each group of skip connections. We show the result of each skip connection switched on the respective swapped group. We observe that the *h-space* has an almost imperceptible effect on the final image, contrary to research into the disentanglement of DDPMs. The first group of skip connections closest to the *h-space* similarly has a limited effect, whereas the most coherent blending occurs in the second group of skip connections. The third group has no coherent effect on the image, generating random distortions, while the fourth performs akin to raw pixel blending.

substitute the skip connection $f_t^l$ of image $A$. We show an example of the effect we obtain by substituting each group of three skip connections (group 1: $l = 1, 2, 3$; group 2: $l = 4, 5, 6$; group 3: $l = 7, 8, 9$; group 4: $l = 10, 11, 12$) and the *h-space* in Fig. 5.

Previous studies [19, 24, 42] indicate that the middle decoding layers or the middle cross- and self-attention blocks are the most determinant of the content, suggesting that the structural information is formed roughly halfway in the decoding blocks. While our method aligns with previous findings, being the third group of skip connections roughly halfway in the depth of the model, it suggests that this information is already encoded in the encoder and passed through the decoder via the residual block.

Accepting standard distinctions of foreground-background and content-style[1], we observe that the injection of the second group of skip connections of image $A$ into image $B$ preserves the background style of image $B$, in this case, the color scheme, the foreground style of image $B$, the stripes of the zebra, the background content of image $A$, the Savannah, and the foreground content of image $A$, the silhouette of the elephant.

### 4.2. The effect of the timesteps

In this section, we investigate the role of timesteps in the diffusion process (see Fig. 7) by injecting the skip connec-

---

[1]While these terms do not have a precise definition, by content, we generally mean the structure of the object, and by style, the colors, textures, and patterns.

Figure 6. Close up into the second group of skip connections. The image shows the effect of this group's different combinations of connections. From the bottom to the top, we injected only one of the skip connections, groups of two, and finally, all three. Specifically, we observe that the combination of $l = 4$ and $l = 5$ carry the most information: $l = 5$ injected alone creates a minimal change in the image but, when combined with $l = 4$, determines the spatial structure of the output. $l = 4$ alone conveys structure only of the foreground.

tion of image $A$ into image $B$ at $t_{start} \neq 1000$ and $t_{end} \neq 0$. We observe that the first 150 steps ($t_{start} = 850$) have little impact on the final image, while the last 150 steps ($t_{end} = 150$) only serve as refinement, as found also in Asyrp [23]. We find that the skip connection of image $A$ or image $B$ for the first 500 denoising steps determines the content of the foreground, while the last 500 steps determine the background.

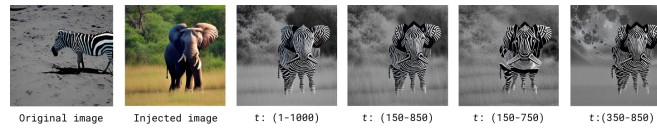

Original image   Injected image   t: (1-1000)   t: (150-850)   t: (150-750)   t:(350-850)

Figure 7. Visualization of the effect of the injection timesteps. We observe that starting the injection later at $t_{start} < 850$ leads to distortions in the foreground content while ending the denoising earlier at $t_{end} > 150$ reveals the background content of the original image. This phenomenon is consistent for every image we generate.

### 4.3. Modulating the effect

To achieve more controllable results, we investigate methods to modulate the intensity of the change.

**Injection classifier-free guidance.** Inspired by classifier-free guidance [15], we test the use of a linear combination of the injected embedding and original embedding of the changed skip connections, parametrized by $\gamma$ to balance the intensity of the mix. At each denoising step, the

injected embedding becomes:

$$f^A(t,l) = f^B(t,l) + \gamma(f^A(t,l) - f^B(t,l)) \quad (2)$$

where $t$ is the denoising timestep and $l$ the skip connection layer.

**Depth-wise alternation of the spatial embedding of the skip connections.** The *h-space* and each skip connection are high-dimensional matrices with depth, width, and height channels. For instance, the layer $l = 4$ for a $512 \times 512$ output size is $1280 \times 16 \times 16$, so $depth = 1280$ and $width = height = 16$, as opposed to traditional latent spaces of GANs and VAEs of size 512. We hypothesize that the information stored in these embeddings is, therefore, highly redundant and attempt to investigate the nature of these spaces' spatial features (width and height). We plot these embeddings as 1280 images of $16 \times 16$ pixels and we find that over $90\%$ of the kernels show the same shape with varying average or inverse intensities. Therefore, we suspect redundancy in the depth channel.

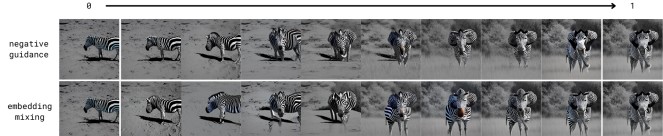

Figure 8. Visualization of modulation methods. We show the effects of the two modulation methods at $\gamma = r \in [0, 1]$. We observe that both methods achieve a successful modulation of the intensity of the effect and empirically observe that the use of both methods together obtains the best results. The advantage of the guidance is that it can surpass the effect above 1, but, differently from the second modulation method, it struggles in areas around 1, where the image should be similar to the non-modulated effect.

We leverage this observation to introduce an additional modulation method: we alternate at a ratio $r$ the kernels of $f^A(t,l)$ with those of $f^B(t,l)$. That is to say, for every $1280 \times r$ $16 \times 16$ kernels, we inject the injection kernel and maintain the original one in the other cases.

In sum, the injection timing can control whether the background is retained or replaced with that of the original image, and the injection strength can be further modulated using classifier-free guidance and depth-channel alternation.

## 5. Experiments

We evaluate our method on **image editing** and **style transfer**, providing both quantitative metrics and qualitative results. To evaluate our method on text-guided image-to-image and text-to-image translation, we follow established benchmarks, utilizing the Wild-TI2I dataset [42] and ImageNet-R-TI2I [42]. We adopt the protocol outlined in

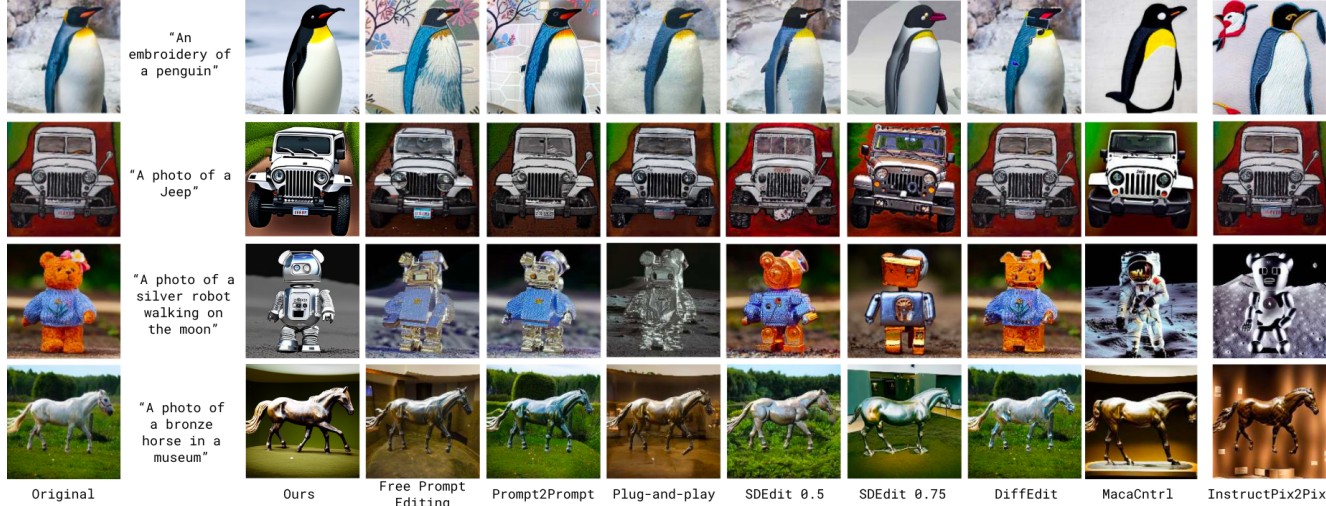

Figure 9. Qualitative comparison of different prompt-guided editing methods. We use as reference results proposed by [24] thus, we do not cherry-pick the results. From left to right: source image, target prompt, our result, Free Prompt Editing, P2P [12], PnP [41], SDEdit [26] with two noise levels, DiffEdit [9], Pix2pixzero [25], Shape-guided [29], MasaCtrl [5], and InstructPix2Pix [3] (a fine-tuning-based method).

[19] for style transfer evaluation to text-guided style transfer.

Our evaluation employs two complementary metrics. First, text-image CLIP similarity quantifies how closely the generated images align with the style or edit prompts [30]. Second, the distance between DINO ViT self-similarity [6] assesses the degree of structure preservation. Additionally, we use LPIPS [47] to measure perceptual similarity, where lower values indicate better content retention.

We implement our method with the `Diffusers` library, using a custom `2DUNetConditional` model based on pre-trained weights from `stabilityai/stable-diffusion-2-base`. For image-to-image translation, we apply the `DDIMInverseScheduler` with 50 steps, generating images with the `UniPCMultistepScheduler` using 50 inference steps and a guidance scale of 7.5.

cd

### 5.1. Image editing

**Qualitative Analysis** provides a comparative analysis with previous methods. Competing methods frequently exhibit issues: Free Prompt Editing lacks style specificity (e.g., "penguin embroidery" fails to capture the embroidery texture), Prompt2Prompt does not follow the prompt effectively (the horse is not in the museum), and Plug-and-Play leads to feature distortions (e.g., "silver robot"). SDEdit struggles with structural integrity at high noise levels, while DiffEdit and MaCaCntrl lose context (e.g., the "teddy bear" is distorted). In contrast, our model consistently delivers prompt-specific transformations with high structural fi-delity, demonstrating robustness across various styles and editing demands.

**Quantitative analysis** Quantitatively, we present the performance of our methods on the ImageNet-R-TI2I and Wild benchmarks. In Fig 10 we evaluate our model with CLIP cosine similarity (indicating prompt fidelity) and DINO-ViT self-similarity (indicating structural preservation). Across all benchmarks (Wild-TI2I, ImageNet-R-TI2I, and Generated ImageNet-R-TI2I), our model consistently balances high CLIP similarity with low DINO self-similarity, outperforming other methods like SDEdit, VQGAN-CLIP, and DiffuseIT in both text alignment and structural accuracy. Notably, our approach consistently places in the "Better" region, reflecting superior text fidelity and structural integrity.

### 5.2. Style Transfer

**Qualitative evaluation** Figure 11 offers a comparative analysis, highlighting distinctive performance variations among competing models. Models like DiffStyler, CLIP-Styler, and Plug-and-Play often compromise the fidelity of the original content structure, leading to blurred or distorted shapes, particularly in intricate or highly abstract styles. NTI+P2P exhibits minimal style alteration, evident in the "8-bit pixel art" transformation, where the ship closely resembles the original. However, it is relevant to note that while these models demonstrate varying degrees of style application, evaluating artistic styles can be inherently arbitrary. Styles intended as artistic movements are sometimes conflated with specific methods, making objective assessment challenging. For instance, applying a "Dadaism style"

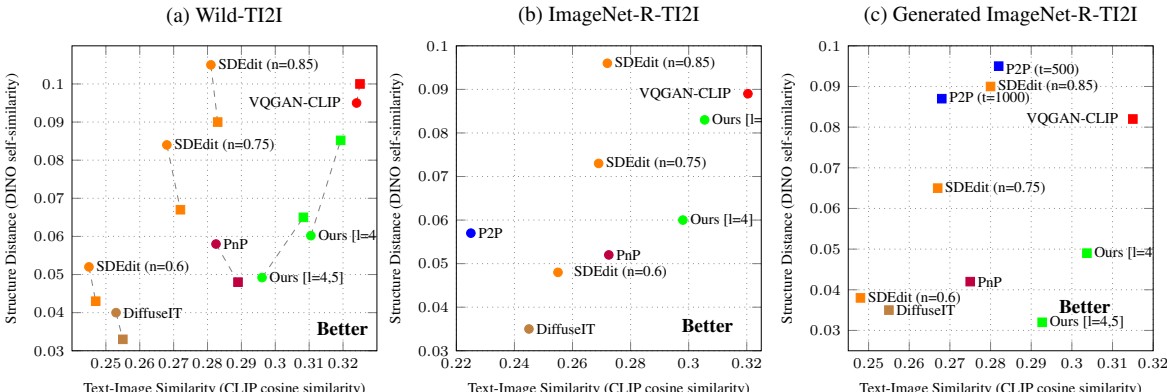

Figure 10. Quantitative evaluation. We measure CLIP cosine similarity (higher is better) and DINO-ViT self-similarity distance (lower is better) to quantify the fidelity to text and preservation of structure, respectively. We report these metrics on three benchmarks: (a) Wild-TI2I, (b) ImageNet-R-TI2I, and (c) Generated ImageNet-R-TI2I. [42]

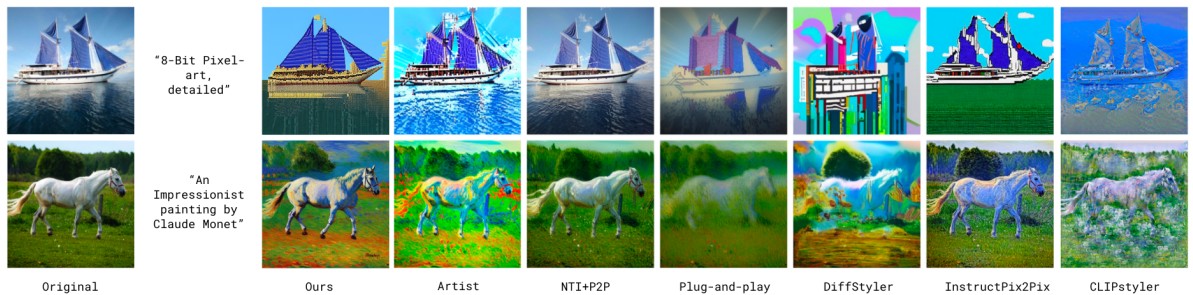

Figure 11. Qualitative evaluation against current style transfer methods. We use the reference results by [19]
, and we do not do any cherry-picking.

prompt may focus on collage techniques rather than capturing the movement's conceptual essence.

In contrast, our model achieves a balanced and coherent output across styles, effectively preserving not only the content structure and the stylistic features but also adjusting the people, clothing, and objects in a historically coherent manner (as in Fig. 11). For example, in the "Impressionist painting" transformation, our model accurately replicates the brushstroke aesthetic and introduces a poppy field, typical of Impressionist painters, while maintaining the original shape and posture of the horse. Nonetheless, our method inherits certain biases from Stable Diffusion, resulting in inaccurate visual aesthetics for movements like Cubism, Futurism, and Dadaism despite successfully achieving stereotypical modifications.

In Fig. 12, we show a practical application of the style transfer features of our model, demonstrating interesting applicability to styles presented by the creative communities adopting Midjourney and StabilityAI using both a prompt and an image for style transfer. Note that the latter has not been shown to work for competing works.

**Quantitative evaluation** Our method, represented by **l=4,5**

and **l=4** in Table 1, demonstrates strong alignment with text prompts while preserving content structure. On the CLIP Alignment metric, the **l=4** model achieves the highest score of 28.55, with **l=4,5** close behind at 26.27. These scores indicate that our model adheres effectively to prompt guidance, achieving transformations that accurately reflect the target style. Regarding structural similarity, our **l=4,5** model attains an LPIPS score of 0.57, with **l=4** following at 0.67, demonstrating good content retention compared to most baseline models. These lower LPIPS values suggest that our approach maintains structural and perceptual fidelity to the original content, even under significant stylistic transformations. Competing methods, such as DDIM (0.74), DiffStyler (0.72), and ControlNet-Depth (0.78), display higher LPIPS scores, reflecting a greater degree of content distortion. Artist shows competing performance while obtaining inferior qualitative results.

Finally, in Fig. 13, we show the impressive editing results achieved on the turbo-distilled version of the model. Previous works have not shown this applicability.

Table 1. Evaluation of Different Style Transfer Models on the Artist Dataset [19], measuring Content Preservation (LPIPS) and Stylization Prompt Alignment (CLIP Alignment).

| Metric | Ours (l=4,5) | Ours (l=4) | Artist | DDIM | NTI-P2P | PnP | DiffStyler | InstructP2P | ControlNet-Canny | ControlNet-Depth | CLIPStyler |
|---|---|---|---|---|---|---|---|---|---|---|---|
| LPIPS ↓ | 0.57 | 0.67 | 0.62 | 0.74 | 0.67 | 0.67 | 0.72 | **0.47** | 0.72 | 0.78 | 0.51 |
| CLIP Alignment ↑ | 26.27 | **28.55** | 28.33 | 28.38 | 25.87 | 26.4 | 26.82 | 23.59 | 26.4 | 27.05 | 26.14 |

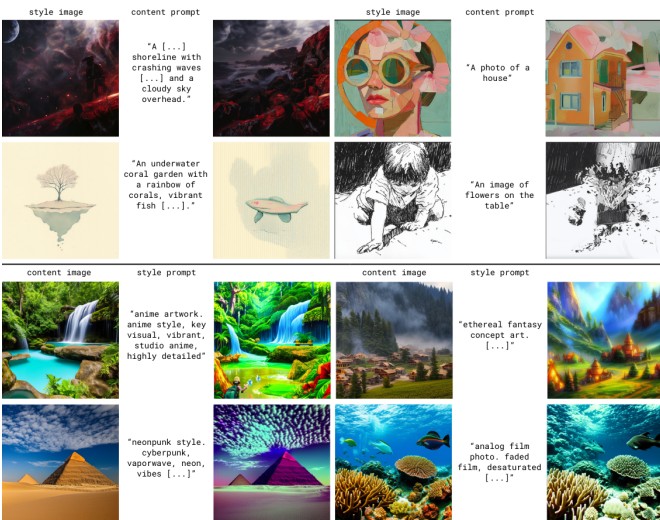

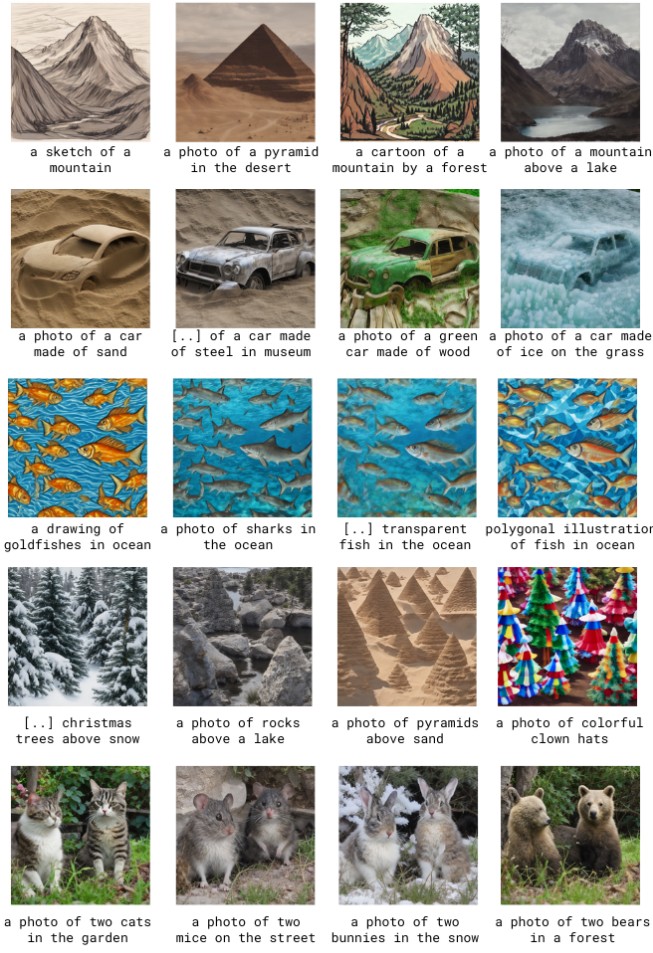

Figure 12. The AI Art online communities offer an incredible wealth of information on style transfer in blogs such as Stable Diffusion Art that could be leveraged to build applied benchmarks for style transfer. In this figure, we show two interesting applications of our method: the first consists of the transfer of closed-source styles (e.g., styles used in Midjourney) to Stable Diffusion outputs using single-image style transfer (on the left). The second leverages the style prompts (with respective negative prompts) released by StabilityAI to transfer the described styles to real images or selected generated images (on the right).

## 6. Conclusion

In conclusion, this paper explores the impact of U-Net skip connections in Stable Diffusion models, presenting a training-free, efficient approach - SkipInject - that enables high-quality text-guided image editing and style transfer. By systematically examining these skip connections, we address key questions about how spatial and stylistic information is encoded in the latent spaces of Stable Diffusion, the stages within the denoising process where they arise, and the structure of these spaces. Our findings reveal that specific skip connections are fundamental in controlling content and style, providing insight into how these components influence image generation.

The proposed method leverages the l=4 and l=5 skip connections to achieve precise style and content transfer, demonstrating state-of-the-art or on-par performance across established benchmarks. In addition, we introduce three modulation techniques for controlled editing intensity, of-

Figure 13. Example results of text-based image editing using Stable Diffusion Turbo with 1 step inference on `wild-ti2i-fake`. The modifications obtained are coherent and cohesive, obtaining radical changes and maintaining the original structure. Compared to multi-step inference, the control over the background is more limited.

fering flexible adjustments to meet diverse requirements.

Our approach currently relies on a single latent, limiting its application from scenarios that require dual-image style transfer. Future work will focus on extending SkipInject to support two-image inputs for broader applications.

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
