# OpenReview forum: "Training-Free Style and Content Transfer by Leveraging U-Net Skip Connections in Stable Diffusion"
_thecvf.com/CVPR/2025/Workshop/CVEU — CVPR 2025_

### Official Review · Reviewer_2Mbu · 2025-03-12

**Rating:** 4
**Confidence:** 4

**Review:**

==== Summary ====

This paper studies the skip-connections in Stable Diffusion's denoising U-Net, a previously overlooked component in pre-trained diffusion models. The authors observe that the skip-connections of the second encoder stage decompose image style from content. By swapping these features of two images, they can transfer the style of the second image to the first image. The proposed method, SkipInject, is applicable to image and text based editing tasks. Experimental results show the superior performance of SkipInject over baselines.

==== Strengths ====

- I enjoy reading Section 4, the analysis on U-Net skip connections. The writing and the reasoning flow is very clear. The results with different U-Net blocks and denoising timesteps are insightful.
- The figures are good. They help illustrate the motivation and the effect of each design choice well.
- Both qualitative and quantitative experimental results are solid.

==== Weaknesses ====

- The method is only applicable to U-Net based diffusion models. This limits its scope and importance given recent strong models are all Transformer-based.
- I am a bit confused by "Depth-wise alternation of the spatial embedding of the skip connections". How do you choose which channels to inject, and which to retain? In addition, there is no ablation about this design.
- Lacking user study results for a more comprehensive comparison with baselines. But I feel this is fine for a workshop paper

---

### Official Review · Reviewer_rue5 · 2025-03-17
**Simple but novel approach, lack of ablation and user studies.**

**Rating:** 3
**Confidence:** 4

**Review:**

The paper introduces a training-free image editing method using U-Net skip connections in Stable Diffusion. The approach is simple but the controllability is effective. The finding of the third encoder block's residual connections playing critical role during image generation is inspiring. Extensive experiments and ablation studies show the effectiveness of the proposed method.

### Strengths:

* The method using residual block for image editing is simple and novel. The findings of the use of residual block are interesting.
* The analysis of results is detailed and comparisons with other methods are sufficient (Figure 9, 10, 11).
* The transferability of this method is good. In addition to the original Stable Diffusion model, this method also benefits from an accelerated version. (Line #394)

### Weaknesses:

* Ablation studies are insufficient, such as influenced skip connections, timesteps and classifier-free guidance. The analysis focuing on visualization of a specific case (e.g., Figure 6, 7) is quite empirical.
* For image editing results, it is better to do sufficient users study since quantitative evaluation is not quite realiable in some circumstances.

---

### Official Review · Reviewer_7GB4 · 2025-03-23
**Analysis is interesting**

**Rating:** 4
**Confidence:** 4

**Review:**

The paper presents a method for image editing and style transfer based on an analysis of the residual connections in Stable Diffusion. This analysis reveals a separation between content and style in the image generation process. Leveraging this separation, the method enables edits that preserve the structure while modifying the appearance, and vice versa. The approach is evaluated both qualitatively and quantitatively against previous image editing and style transfer methods.

Overall, I find the analysis insightful and the results compelling.

One question I have regarding the results is why the background appearance is well preserved in some cases but significantly altered in others.

Additionally, the description of the style transfer method could be more explicit. Including pseudo-code or a figure illustrating each of the methods would likely improve clarity.

Another concern relates to the transferability of the insights in this paper to more advanced architectures. Since SDXL is also based on a U-Net, it would be interesting to see if the method generalizes to it.

---

### Official Review · Reviewer_C66M · 2025-03-23
**Training-Free Style and Content Transfer by Leveraging U-Net Skip  Connections in Stable Diffusion**

**Rating:** 3
**Confidence:** 4

**Review:**

This paper investigates the role of skip connections in the U-Net architecture within Stable Diffusion, analyzing their properties, their influence on the generated image, and their variations across different time steps. Furthermore, based on these findings, the paper proposes an efficient and controllable image editing method. Experimental results demonstrate the effectiveness of the proposed approach. However, several concerns and questions remain regarding this paper.

1.	In Section 3.2, the author states that the proposed method can be applied to Stable Diffusion versions 1.4, 1.5, 2, and 2.1. I'm curious whether this method can also be applied to SDXL. SDXL is also based on the U-Net architecture.
2.	The role of skip connections was initially explored in the FreeU paper. However, FreeU primarily applies them to enhance text-to-image and text-to-video generation. I'm particularly interested in understanding the fundamental differences between this paper and FreeU.
3.	The comparison methods chosen in this paper are from 2023 and earlier, lacking a comparison with more recent approaches. While it's not necessary to completely surpass the latest methods, it would be valuable to understand the advantages of this paper's approach compared to them.

---

### Decision · Program_Chairs · 2025-03-25

**Decision:**

Accept

**Comment:**

The paper introduces a training-free method, SkipInject, leveraging skip connections in Stable Diffusion's U-Net architecture for controllable image editing and style transfer. Reviewers praised the insightful analysis of skip connections, clear writing, compelling qualitative and quantitative results, and practical effectiveness. However, reviewers noted some limitations, such as applicability mainly to U-Net architectures, insufficient ablation studies, lack of user studies, and limited comparisons with recent methods.

Given the overall positive feedback and the method's demonstrated effectiveness, the paper is accepted. Authors are encouraged to address reviewers' concerns by clarifying the method's applicability to newer architectures (e.g., SDXL), expanding ablation studies, and considering user evaluations in the camera-ready submission.